# Crude Saponins from *Chenopodium quinoa* Willd. Reduce *Fusarium* Wilt Infection in Tomato Seedlings

Xueyong Zhou *, Huan Guo, Lihong Zhang, Liyan Yang, Zuofu Wei, Xiaoying Zhang and Yan Niu

School of Life Science, Shanxi Engineering Research Center of Microbial Application Technologies, Shanxi Normal University, Taiyuan 030000, China; guohuansx@163.com (H.G.); zhangyongyuan0@163.com (L.Z.); yangswallow163@163.com (L.Y.); zorff@163.com (Z.W.); zhangxiaoying0217@163.com (X.Z.); niuyan1010@foxmail.com (Y.N.)
* Correspondence: zhouxueyongts@163.com

**Abstract:** Quinoa saponins are pentacyclic triterpene compounds composed of one triterpenoid glycoside and two different sugar chains. Previous studies have showed that natural quinoa saponins showed little or no antifungal activity, and there are few reports about their antifungal effects in recent decades. *Fusarium* wilt caused by *Fusarium oxysporum* f. sp. *lycopersici* (FOL) is the most serious for tomatoes in the field and under greenhouse conditions. The main objective of this study was to investigate the effectiveness of different concentrations and application modes of crude saponins from quinoa bran against the causal pathogen of tomato wilt under a greenhouse experiment. The results showed that the anti-FOL activity of quinoa saponins was weak in vitro, but significantly enhanced in vivo. Tomato seeds and seedlings treated with solution of quinoa saponins at 0.5 and 1.0 g/L significantly reduced the disease incidence (%) of tomato *Fusarium* wilt. The treatment types of saponin solution have influence on the preventive effects (%) of tomato seedlings against *Fusarium* wilt, among them, root soaking > foliar spray > seed soaking. The treatment of seed soaking with quinoa saponins inhibited germination of tomato seeds to some extent. However, the germination rate of tomato seeds after saponin soaking was comparable to the chemical pesticide (thiram carboxin); therefore, it could be used to control tomato wilt disease. This is due to the fact that the antifungal activity of quinoa saponins in vivo was much higher than that in vitro when the saponin concentration was between 0.5–1.0 g/L, indicating that the antifungal activity of quinoa saponins may be achieved mainly by inducing resistance. This investigation supports the potential use of quinoa saponins as a supplier of antifungal compounds, and could be the foundation for a future study examining the use of quinoa bran as a new resource against FOL.

**Keywords:** tomato *Fusarium* wilt; antifungal activity; quinoa saponins

## 1. Introduction

Tomato (*Solanum lycopersicum* L., Solanaceae) is one of the most profitable and popular fruit or vegetable crop grown worldwide, and it is ranked the second after the potato (*Solanum tuberosum* L.) among vegetable plants [1,2]. Currently, the global production of fresh tomato fruit has reached 187 million tons, and the cultivated area is approximately 5 million hectares [3]. China is the world's largest producer, with a cultivated area of almost 1.11 million hectares [4]. Tomato crops are susceptible to several diseases, which reduce their production. The main diseases in tomato aerial parts are gray mold (*Botrytis cinerea*) and *Cercospora* leaf mold (*Cercospora fuliginea*) [5]; however, the main diseases in tomato underground parts are destructive vascular wilts, rots, and damping-off diseases [6]. Of these soil-borne diseases, tomato wilt caused by *Fusarium oxysporum* f. sp. *lycopersici* (Sacc.) W. C. Snyder and H. N. Hans (FOL) is the most serious in the field and under greenhouse conditions [2,7]. FOL produce three types of asexual infectious spores: macroconidia, microconidia, and chlamydospores [8]. The pathogens of *Fusarium* wilt mainly spread in

farm tools, seeds and soil in the form of mycelium, conidia and chlamydospore [9]. There are three different physiological races of FOL, resulting in its strong pathogenicity and variability [10].

FOL usually infects the roots of tomato via penetration or via wounds, and colonizes in the vascular tissue, resulting in the obstruction of water and nutrient transport [11]. At the beginning of the disease, only one side of the stem appears concave from the bottom to the top, from the oldest to youngest leaves, causing the leaves on one side of the stem to turn yellow. The typical symptoms caused by FOL pathogens include the stunting of seedlings, yellowing and defoliation of leaves, browning of vascular tissues, and eventual death of the whole plant (within about 15–30 d) [12,13]. One of the biotic stresses caused by FOL is responsible for intense yield losses of tomato. According to reports in the literature, FOL causes yield losses in a range between 20% and 80% [3,14,15]. As far as the cherry tomato was concerned, about 65% yield loss and 50–60% plant mortality were observed after infection by FOL [16]. The disease management of tomato caused by FOL mainly includes (1) chemical control; (2) physical control; (3) agricultural control; (4) breeding control; and (5) biological control [17]. The major measure in chemical control is the use of chemical fungicides, which pose dangerous problems to human, animals, and the environment [18]. Physical control measures are mainly achieved through soil improvement. Since FOL is a soil-borne plant pathogen, these fungal pathogens in soil can be killed in a closed greenhouse or shed at high temperature [19]. However, the enclosed shed operation requires suitable climatic conditions. Moreover, it has the disadvantages of cumbersome operation and high cost. Agricultural control is the traditional method for controlling FOL; the main strategies include (1) crop rotation; (2) deep tillage for lime-mixed soil; and (3) the use of organic fertilizers. Although traditional agricultural measures can reduce the occurrence of wilt diseases, they have failed to effectively suppress *Fusarium* wilt [12,20]. The goal of breeding control is to breed varieties of tomato that are resistant to FOL [21]. There is no doubt that the use of resistant cultivars against *Fusarium* wilt is a viable option; however, the occurrence and development of new pathogenic races is a continuous problem [22]. Due to the variability in pathogenicity, the common strategy of pathogens is to co-evolve with resistant cultivars and eventually overcome plant resistance, reflecting the difficulty of controlling them via traditional breeding techniques. Difficulties in FOL control and people's concern for the environment are a strong reason for alternative ecofriendly methods of disease control. Biological control offers a promising alternative to manage this disease due to its ecofriendly nature compared to chemical fungicides. So far, a variety of beneficial microorganisms have been used as biological control agents for tomato wilt. Among them, bacterial control agents include *Acinetobacter* spp. [7], *Bacillus* spp. [23], *Pseudomonas* spp. [24], *Serratia* spp. [15,25]; fungal control agents include *arbuscular mycorrhizal* fungi (AMF) [26], *Serendipita herbamans* [11], *Trichoderma harzianum* [27], *Penicillium oxalicum* [28], etc. In addition, some plant extracts have good control effect on tomato wilt. According to recent reports, the extracts from *Origanum vulgare* L. [29], *Thymus* genus [30], and *Calotropis procera* [1] reduced the incidence of *Fusarium* wilt in tomato seedlings or reduced the pathogen growth. Compared to microbial preparations, the active substances extracted from plants such as essential oils have the advantages of high yield and low cost. Therefore, it is of great significance to explore new plant sources for resisting tomato wilt.

Quinoa saponins are pentacyclic triterpene compounds composed of one triterpenoid glycoside and two different sugar chains [31]. The saponins are mainly located in the outer layers of quinoa, which generate a bitter taste [32]. Although the husk of quinoa accounted for only 10–12% of the seed weight [33], it contained 86% of the total saponins [34,35]. Therefore, saponins can be extracted from the quinoa seeds. Quinoa bran is an industrial by-product of quinoa processing; however, it cannot be used as feed directly due to its high content of saponins, so the utilization of saponins from industrial wastes is another pro-ecological aspect of "zero waste". Previous studies have shown that natural quinoa saponins (both bidesmosides and derived monodesmosides) showed little or no antifungal

activity (*Candida albicans*) [36]. Stuardo and Martín [37] reported that untreated quinoa saponins showed minimum activity against the mycelial growth of *Botrytis cinerea*. However, when the saponin extracts were treated with alkali, their antifungal activity against *B. cinerea* was significantly enhanced. The above results indicated that quinoa saponins had the potential of antifungal activity, which stimulated our research interests.

The objectives of this study were (1) to investigate the effectiveness of different concentrations and application modes of crude saponins from quinoa bran against the causal pathogen of tomato wilt within a greenhouse experiment; (2) to investigate the effect of quinoa saponins on the seedling emergence of tomato; and (3) to evaluate their possible potential for use as alternative bio-pesticides against FOL.

## 2. Materials and Methods

### 2.1. Extraction of Saponins from Quinoa Bran

Quinoa bran was provided by Shanxi Yilong Quinoa Development Limited Company, Jingle County, Shanxi Province. The extraction of crude saponins from quinoa bran was carried out according to Dong et al. [38]. Briefly, 100 g of quinoa bran powder was soaked in ethanol solution (75%, *w/w*) for 2 h at a ratio of 1:15. After ultrasonic extraction for 30 min, the suspensions were centrifuged at 8000× *g* for 15 min, and the supernatants were obtained. The saponins in supernatants were extracted three times with *n*-butanol and combined. After rotary evaporation and freeze-drying, the crude saponins were obtained [39]. The content of saponins was 66.1% by UV−Vis spectrophotometry (Table 1).

**Table 1.** Proximate composition of crude saponins from quinoa bran.

| Saponin (%) | Protein (%) | Fat (%) | Polyphenols (%) | Flavonoids (%) | Tannins (%) | Moisture (%) | Ash (%) |
|---|---|---|---|---|---|---|---|
| 66.1 ± 0.4 | 5.97 ± 0.15 | 0.45 ± 0.11 | 0.25 ± 0.01 | 0.61 ± 0.10 | 0.57 ± 0.21 | 5.77 ± 0.04 | 9.5 ± 1.0 |

### 2.2. Proximate Analysis of Crude Saponins from Quinoa

The saponin content in crude saponins was found using the method of Zhou et al. (2023) [40]. The saponin extract (100 mg) was dissolved in distilled water (100 mL), then the saponin solution (0.5 mL) was transferred into a test tube and mixed with 0.5 mL vanillin–anhydrous ethanol solution (8%) and 4 mL sulfuric acid solution (77%). After mixing evenly, the mixtures were heated at 60 °C in a water bath for 30 min. The test tube was taken out and put into an ice bath to terminate the color reaction. The absorbance of the sample solution was determined at 535 nm by using an ultraviolet-visible spectrophotometer (TU-1810, Beijing Puxi General Instrument Co., Ltd., Beijing, China). The total polyphenol, flavonoid and tannin contents of crude saponins were measured according to the method of Hirose et al. (2010) [41], Sharma et al. (2022) [42], and Zhou et al. (2023) [40], respectively. The protein content of crude saponins was determined by the Kjeldahl method, and the amount of protein was calculated as $6.25 \times N$ (Nitrogen content). Fat was measured in a Soxtec system by extraction with petroleum ether (with a boiling range of 30–60 °C). The total ash content was determined by the Muffle furnace burning method at $850 \pm 25$ °C for 4 h. The moisture content was determined by drying method at $105 \pm 2$ °C for 2 h.

### 2.3. Effects of Soaking Seeds with Saponin Solutions on Tomato Seedling Emergence

Quinoa crude saponins were dissolved in distilled water and prepared into 0.5 g/L and 1.0 g/L solutions, respectively. Fifty tomato seeds were soaked in each concentration of saponins for 30 min, and then the seeds were taken out to dry naturally. The distilled water was used as the control. Each treatment (50 seeds) was repeated four times, and the treated seeds were planted in a nutritional box filled with turf, vermiculite, and vegetable soil (*w:w:w* = 2:1:1). The seedling emergence of tomato was investigated on the seventh and

tenth day after sowing, respectively. The number of tomato seedlings was recorded, and the seedling emergence rate was calculated using the following Equation (1):

$$\text{Seedling emergence rate (\%)} = (\text{Seedling number}/\text{seeding number}) \times 100 \qquad (1)$$

### 2.4. FOL Strains and Growth Conditions

FOL was isolated and identified by the Plant Disease Laboratory of Tianjin Institute of Plant Protection, China. Before the experiment, FOL (pathogen) was grown on PDA (potato dextrose agar medium) plates at 25 °C for 7 days for routine cultivation. During in vitro antifungal activity tests of quinoa saponins, the mycelial cake of FOL in Petri dishes was used as an inoculum (see Section 2.5). The conidial suspensions were prepared by flooding the plates with sterile water containing 0.05% (*v/v*) Tween-80, and gently scraping the agar surface (the mycelium) with a sterile scalpel blade. The suspension was filtered through two layers of sterilized gauze, and the conidia were enumerated with a hemocytometer and adjusted to a final concentration of $1 \times 10^6$ spores/mL. The conidial suspensions were used for inoculation when tomato seedlings were transplanted (see Section 2.7).

### 2.5. Preparation of FOL Soil Inoculants

Grain sorghum was soaked in tap water at room temperature for 24 h, then sterilized by autoclave at 121 °C for 30 min. After cooling to room temperature, the grain sorghum was inoculated with conidia suspension of FOL and incubated at 25 °C for two weeks. The grain sorghum with fungus was mixed with the planting soil at 3% (*v/v*). The soil infected by FOL was used for inoculation when tomato seedlings were transplanted (see Section 2.7).

### 2.6. Effect of Quinoa Saponins on the Linear Mycelial Growth of FOL In Vitro

The antifungal activity of quinoa saponins against FOL in vitro was evaluated by the mycelial growth rate method. Briefly, 2.5 and 5 g of quinoa saponins were dissolved in 1000 mL sterile water, respectively, and the concentration of quinoa saponins was 2.5 and 5 g/L, respectively. Ten milliliters of saponin solution was added to 40 mL melted PDA medium (containing 100 mg/L streptomycin sulfate) and mixed thoroughly (making the total volume 50 mL), in which the concentration of saponins was 0.5 and 1.0 g/L, respectively. Ten milliliters of sterile water were used as blank control. Each 50 mL of the mixed medium was poured into six sterilized Petri dishes (diameter 9 cm). A circle of 4 mm diameter was then removed from the edges of the vigorously developing FOL colonies and moved to the middle of the solidified PDA medium [43]. All inoculated plates were stored in an incubator at 25 °C. The increased value of the colony diameter was measured at 48 and 72 h, respectively. The inhibition rate of mycelial growth was calculated using Equation (2).

$$\text{Inhibition rate of mycelial growth (\%)} = \frac{A - B}{A} \times 100 \qquad (2)$$

where A is the increased value of the colony diameter in the control group, mm; B is the increased value of the colony diameter in the treatment group, mm.

### 2.7. In Vivo Antifungal Activity of Seedlings after Seed Treatment

As described in Section 2.3, most seeds treated with quinoa saponins can grow into seedlings. When there were 3–4 true leaves on tomato seedlings, these seedlings were ready for transplanting into pots in a greenhouse. Night and day temperatures ranged from 15 to 25 °C, the relative humidity was between 60% and 80%, and the lighting time was 12 h. For FOL inoculation, tomato seedlings were uprooted from the nutritional medium, and roots were immersed in a conidial suspension ($1 \times 10^6$ spores/mL) for 60 min [2]. The inoculated plants were transferred to the pots (the height of which was 7.5 cm, and the inner diameter was 9 cm) filled with the infected soil (containing 3% sorghum, see Section 2.5: Preparation

of FOL solid inoculants). Each treatment had 35 seedlings, and four replicates were used. Disease incidence of seedlings was investigated after 10 days of infection.

The disease severity was recorded on a scale of 0–4, with 0 representing no infection and 4 denoting plants completely infected [5]. The 0–4 scale of disease severity was evaluated by determining the proportion of wilted leaves using the following method [2]:

0: No wilt symptoms;
1: Slight infection, which is about 1% to 33% of leaves being yellowed or wilted;
2: Moderate infection, which is about 34% to 67% of leaves being yellowed or wilted;
3: Extensive infection, which is about 68% to 100% of leaves being yellowed or wilted;
4: Complete infection, which is the whole plant wilting, and plant death.

The percentage of disease incidence was determined using Equation (3).

$$\text{Disease incidence } (\%) = \left[\frac{\sum \text{scale} \times \text{number of plants infected}}{(\text{highest scale} \times \text{total number of plants})}\right] \tag{3}$$

$$\text{Preventive effects } (\%) = \left[\frac{C - D}{C}\right] \times 100 \tag{4}$$

where C is the disease incidence of the control, %; D is the disease incidence of saponin treatment, %.

### 2.8. In Vivo Antifungal Activity of Seedlings after Root Soaking Treatment

Quinoa seeds (50) without saponin treatment were planted in nutritional box filled with turf, vermiculite, and vegetable soil (w:w:w = 2:1:1). When there were 3–4 true leaves on tomato seedlings, these seedlings were uprooted from the nutritional medium, and roots were soaked in a solution of quinoa saponins for 5 min. The concentration of saponin solution was 0.5 and 1.0 g/L, respectively. After taking out from the solutions, these seedlings were placed on a shelf with holes for about 2 h. Then, the roots of seedlings were immersed in a conidial suspension ($1 \times 10^6$ spores/mL) for 60 min. The inoculated seedlings were transferred to the pots (the height is 7.5 cm, inner diameter is 9 cm) filled with the infected soil (containing 3% sorghum, see Section 2.5). Each treatment had 35 seedlings, and four replicates were used. Disease incidence of seedlings was investigated after 10 days of infection. The disease incidence and preventive effects were calculated as described in Section 2.7.

### 2.9. In Vivo Antifungal Activity of Seedlings after Foliar Spray Treatment

Quinoa seeds (50) without saponin treatment were planted in nutritional box filled with turf, vermiculite and vegetable soil (w:w:w = 2:1:1). When there were 3–4 true leaves on tomato seedlings, the leaves of seedlings were sprayed with a solution of quinoa saponins whose concentration was 0.5 and 1.0 g/L, respectively. One day later, these seedlings (35 plants) were uprooted, and the roots were immersed in a conidial suspension ($1 \times 10^6$ spores/mL) for 60 min. The inoculated seedlings were transferred according to Section 2.8. The disease incidence and preventive effects were calculated as described in Section 2.7.

### 2.10. Statistical Analysis

All experiments were performed in four replicates. Data were analyzed using a one-way analysis of variance (ANOVA) with SPSS software (IBM Corporation, Armonk, NY, USA). Results were reported as means $\pm$ standard deviation (SD). The significance level is $p < 0.05$.

## 3. Results

### 3.1. Proximate Composition of Crude Saponins from Quinoa

The results of proximate composition of crude saponins from quinoa are shown in Table 1. The saponin content in crude saponins was 66.1%, which was close to that reported

by others [44] (the saponin purity was over 60%). The non-saponin compounds were proteins, fat, polyphenols, flavonoids, tannins, moisture, and ash, which were basically similar to those within the results of Stuardo and Martín [37] (proteins, fat, fiber, and ash). Nevertheless, the ash content of quinoa saponins was much higher than that of quinoa bran [40].

### 3.2. Effects of Seed Soaking with Saponins on Tomato Seedling Emergence

The effects of different concentrations of quinoa saponins on the seedling emergence rate of tomato seeds are shown in Table 2. Some 7 days after sowing, the seedling emergence rate of tomato seeds treated with quinoa saponins ranged from 32% to 40%, which was significantly higher than that treated with a thiram carboxin suspension (1.0 g/L). Furthermore, there were no significant differences between saponin concentrations (0.5 and 1.0 g/L).

**Table 2.** Effect of saponin treatment on tomato seedling emergence rate.

| Treatment Agent | Concentration (g/L) | Seedling Emergence Rate (%) | |
| --- | --- | --- | --- |
| | | 7th Day after Sowing | 10th Day after Sowing |
| Quinoa saponins | 0.5 | 32 [b] | 76 [b] |
| Quinoa saponins | 1.0 | 40 [b] | 72 [b] |
| Thiram carboxin | 1.0 | 12 [c] | 76 [b] |
| Control | 0 | 84 [a] | 96 [a] |

Superscript (a–c) indicates significant differences ($p < 0.05$) in the same column.

Some 10 days after sowing, the seedling emergence rate of tomato seeds was much higher than that on 7th day (Table 2). Compared to the control, the treatment of saponins and thiram carboxin significantly reduced the seedling emergence rate. Nevertheless, the seedling emergence rate of tomato seeds treated with saponins was comparable to that of seeds treated with thiram carboxin, indicating that this effect is within acceptable limits, and quinoa saponins can be used as a potential fungicide.

### 3.3. In Vitro Activity of Quinoa Saponins against FOL

The quinoa saponins were diluted to 0.5 and 1.0 g/L, respectively, and the in vitro antifungal action of quinoa saponins against FOL was observed.

The inhibition rate of quinoa saponins against FOL was between 4.76% and 9.52% (Table 3). Moreover, the antifungal activity in vitro increased with the increase in saponin concentration. As shown in Figure 1, the quinoa saponins had a weak inhibition rate against FOL in vitro when the concentration of quinoa saponins ranged from 0.5 to 1.0 g/L.

**Table 3.** In vitro activity of quinoa saponins against FOL.

| Treatment Agent | Concentration (g/L) | Mycelial Inhibition Rate (%) |
| --- | --- | --- |
| Quinoa saponins | 0.5 | 4.76 ± 0.91 |
| Quinoa saponins | 1.0 | 9.52 ± 1.29 |

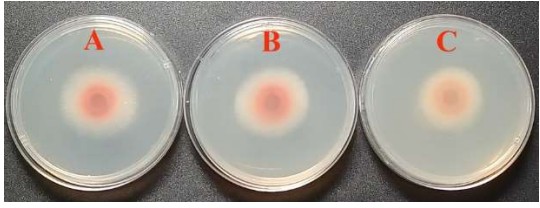

**Figure 1.** Antifungal activity of quinoa saponins against FOL in vitro. ((**A**): control; (**B**,**C**) represent saponin concentrations of 0.5 and 1.0 g/L, respectively).

*3.4. In Vivo Activity of Quinoa Saponins against FOL*

3.4.1. Effect of Seed Soaking with Saponins on Seedling Resistance to Fusarium Wilt

To verify the efficacy of quinoa saponins against *Fusarium* wilt, a greenhouse experiment was performed to evaluate the biocontrol ability of quinoa saponins. When the seed soaking method was used, the disease incidence (%) of tomato *Fusarium* wilt decreased significantly after treatment with quinoa saponins with thiram carboxin (ranged from $27.20 \pm 0.71$ to $32.59 \pm 0.71$) compared with water treatment (Table 4). With the increase in saponin concentration from 0.5 to 1.0 g/L, the disease incidence (%) reduced from $32.59 \pm 0.71$ to $30.58 \pm 0.78$. However, there was no significant difference between concentrations (0.5 and 1.0 g/L) for quinoa saponins.

**Table 4.** Effect of seed soaking with saponins on seedling resistance to *Fusarium* wilt.

| Treatment Agent | Concentration (g/L) | Disease Incidence (%) | Preventive Effects (%) |
|---|---|---|---|
| Quinoa saponins | 0.5 | $32.59 \pm 0.71$ [b] | $49.26 \pm 1.10$ [c] |
| Quinoa saponins | 1.0 | $30.58 \pm 0.78$ [bc] | $52.39 \pm 1.21$ [b] |
| Thiram carboxin | 1.0 | $27.20 \pm 0.71$ [c] | $57.66 \pm 1.10$ [a] |
| Control | 0 | $64.23 \pm 2.61$ [a] | — |

Superscript (a–c) indicates significant differences ($p < 0.05$) in the same column.

After treatment with thiram carboxin, the preventive effects (%) of tomato seedlings against *Fusarium* wilt was 57.66%, while after treatment with quinoa saponin, the preventive effects (%) of tomato seedlings against *Fusarium* wilt ranged from 49.26% to 52.39%. There were significant differences between thiram carboxin and quinoa saponins.

3.4.2. Effect of Root Soaking with Saponins on Seedling Resistance to Fusarium Wilt

When the seedling root soaking method was used, the disease incidence (%) of tomato wilt treated with water was $48.06 \pm 1.83$, while the disease incidence (%) of tomato *Fusarium* wilt treated with quinoa saponins was between $18.24 \pm 0.51$ and $20.61 \pm 0.65$ (Table 5), which was lower than that of the seed soaking method. The disease incidence (%) of tomato seedlings treated with thiram carboxin was the lowest ($16.60 \pm 0.30$), which was significantly lower than that treated by quinoa saponins.

**Table 5.** Effect of root soaking with saponins on seedling resistance to *Fusarium* wilt.

| Treatment Agent | Concentration (g/L) | Disease Incidence (%) | Preventive Effects (%) |
|---|---|---|---|
| Quinoa saponins | 0.5 | $20.61 \pm 0.65$ [b] | $57.12 \pm 1.36$ [c] |
| Quinoa saponins | 1.0 | $18.24 \pm 0.51$ [bc] | $62.04 \pm 1.06$ [b] |
| Thiram carboxin | 1.0 | $16.60 \pm 0.30$ [c] | $65.45 \pm 0.63$ [a] |
| Control | 0 | $48.06 \pm 1.83$ [a] | — |

Superscript (a–c) indicates significant differences ($p < 0.05$) in the same column.

After treatment with thiram carboxin, the preventive effect (%) of tomato seedlings against *Fusarium* wilt was 65.45%, which was higher than that after treatment with quinoa saponins. After treatment with quinoa saponins, the preventive effect (%) of tomato seedlings against *Fusarium* wilt ranged from 57.12% to 62.04%, and there was a significant difference between saponin concentrations (0.5 and 1.0 g/L) ($p < 0.05$). Compared with Table 4, the preventive effect (%) of the root soaking treatment was higher than that of the seed soaking treatment.

3.4.3. Effect of Foliar Spray with Saponins on Seedling Resistance to Fusarium Wilt

When the seedling foliar spray method was used, the disease incidence (%) of tomato *Fusarium* wilt treated with water was $50.35 \pm 0.48$, while the disease incidence (%) of tomato *Fusarium* wilt treated with quinoa saponins was between $22.19 \pm 0.27$ and $23.39 \pm 0.55$, which were both higher than that of the root soaking treatment (Table 6), but lower than

that of the seed soaking treatment. There was a significant difference between saponin concentrations (0.5 and 1.0 g/L) ($p < 0.05$). The disease incidence (%) of tomato seedlings treated with thiram carboxin was the lowest (15.91 ± 0.03), which was significantly lower than that of seeds treated with quinoa saponins ($p < 0.05$).

**Table 6.** Effect of foliar spray with saponins on seedling resistance to *Fusarium* wilt.

| Treatment Agent | Concentration (g/L) | Disease Incidence (%) | Preventive Effects (%) |
|---|---|---|---|
| Quinoa saponins | 0.5 | 23.39 ± 0.55 [b] | 53.54 ± 1.08 [c] |
| Quinoa saponins | 1.0 | 22.19 ± 0.27 [c] | 55.92 ± 0.54 [b] |
| Thiram carboxin | 1.0 | 15.91 ± 0.03 [d] | 68.40 ± 0.06 [a] |
| Control | 0 | 50.35 ± 0.48 [a] | — |

Superscript (a–d) indicates significant differences ($p < 0.05$) in the same column.

After treatment with a foliar spray of thiram carboxin, the preventive effect (%) of tomato seedlings against *Fusarium* wilt was 68.40%, while after treatment with a foliar spray of quinoa saponins, the preventive effect (%) of tomato seedlings against *Fusarium* wilt ranged from 53.54% to 55.92%, which was lower than that of seeds given the root soaking treatment, but higher than those given the seed soaking treatment. There was a significant difference between quinoa saponin concentrations (0.5 and 1.0 g/L) ($p < 0.05$).

## 4. Discussion

*Fusarium* wilt of tomato is one of the most important diseases caused by FOL. If plants are severely infected with *Fusarium*, it can cause permanent wilting and death [3]. Currently, many chemical fungicides are being used to control the disease. However, due to the adverse effects of chemical pesticides on health and the environment, researchers are in search of alternatives and environmentally safe methods for management of plant diseases. Among biological control methods, plant extracts used as biopesticides are one of the most important means. According to reports in recent years, a range of plant extracts can be used to control tomato wilt. For example, essential oil from *Thymbra* [30] and *Origanum vulgare* L. [29]; aqueous extracts from *Calotropis procera* [1], *Solanum linnaeanum* L. [45], *Pistacia lentiscus* (L.) [46], pomegranate peel [47], neem and willow [48], *Curcuma longa* Val., *Allium sativum* L., and *Zingiber officinale* Rosc [49]; methanolic extracts from *Monsonia burkeana* and *Moringa oleifera* [50]; ethanol extracts from *Juglans* spp. and *Carya* sp. [51]; hexane extracts from *Stevia rebaudiana* [52]; ethyl acetate extracts from *Albizia lebbeck* [53]; and chloroform extracts from *Piper betle* L. [54], etc. However, the application of saponins extracted from plants to tomato wilt control is rarely reported. This study aimed to investigate the effects of quinoa saponins on the tomato seedling emergence and biocontrol of tomato *Fusarium* wilt in vitro and in vivo. The results showed that quinoa saponins could inhibit the germination of tomato seeds to some extent, which was largely consistent with Rongai et al.'s results [47] that the aqueous extract of pomegranate peel inhibited germination when the concentration was more than 1.5%. Smaili et al. [55] reported that the semisynthesized triterpene derivatives inhibited tomato germination at higher concentrations (more than 100–500 mg/mL). Nevertheless, Pérez et al. [56] found that triterpenoid saponins from the aerial parts of *Trifolium argutum* Sol. did not affect the tomato germination and shoot development in general. The present study also showed that the effect of quinoa saponins on tomato seedling emergence gradually decreased with the extension of germination time (from the 7th to 10th day, Table 2), and the extent of these effects is comparable to that of a chemical pesticide (thiram carboxin), meaning that this effect is within acceptable limits.

After treatment of tomato seeds and tomato seedlings with quinoa saponins, the disease incidence was significantly reduced compared with the control. When tomato seedlings were treated with quinoa saponin solution through root soaking and foliar spray, the disease incidence was between 18.24–20.61%, and 22.19–23.39%, respectively, which was lower than that of seeds treated with neem and willow aqueous extracts (the disease

incidence of tomato seedlings was between 25.5% and 27.8%), indicating that quinoa saponins may have higher resistance to tomato *Fusarium* wilt in vivo than neem and willow extracts. It was also observed that the treatment modes of saponins had an effect on the resistance to FOL; among them, resistance to FOL was the highest after the root soaking treatment, and the lowest after the seed soaking treatment. The reason for this phenomenon may be that tomato wilt is a soil-borne disease [6], and root soaking treatment can effectively prevent the invasion of FOL. Although the seed soaking treatment can also prevent the invasion of FOL, the saponins may degrade, or the concentration of saponins may decrease after a long growth stage.

Previous studies have shown that the saponin content in plants is consistent with disease resistance, as tested by *Fusarium* inoculation, and the total saponin content could be used as an indicator of *Fusarium* resistance in *Lilium* species [57]. The present study showed that natural quinoa saponins had weak antifungal activity at low concentrations (0.5–1.0 g/L) in vitro, which agrees with the results of Woldemichael and Wink [36]. However, alkaline treatment can increase the biological activity of quinoa saponins due to the formation of hydrophobic saponin derivatives that may have a higher affinity with sterols present in cell membranes [37]. It is very interesting to note that the antifungal activity of quinoa saponins at low concentrations (0.5–1.0 g/L) in vivo was significantly improved compared to that in vitro, indicating that the antifungal activity of quinoa saponins may be achieved mainly by inducing resistance. Ben-Jabeur et al. [30] reported that thyme essential oil is an inducer against gray mold and *Fusarium* wilt in tomato. Aqueous extracts from neem and willow can induce the antioxidant defensive enzymes in tomato seedlings [48]. Recently, Righini et al. [3] reported that water-soluble polysaccharides from *Ecklonia maxima* are a growth promoter of tomato seedlings and an inducer of resistance to Fusarium wilt. Therefore, the mechanism of quinoa saponins' resistance to tomato wilt disease in vivo may be different from that proposed by Stuardo and Martín [37], which deserves further study. In addition, the crude saponins of quinoa contain polyphenols, flavonoid, tannins, which can inhibit the growth of microorganisms or have antimicrobial potential [58,59]. Whether these substances have synergistic effects on the antifungal activity of saponins is also worth further study.

## 5. Conclusions

Tomato seeds and seedlings treated with a solution of quinoa saponins at 0.5 and 1.0 g/L significantly reduced the disease incidence (%) of tomato *Fusarium* wilt. After treatment with quinoa saponins through seed soaking, root soaking, and foliar spray, the preventive effect of tomato seedlings against to FOL was between 49.26–52.39%, 57.12–62.04%, and 53.54–55.92%, respectively. The treatment types of saponin solution have an influence on the preventive effects (%) of tomato seedlings against *Fusarium* wilt, among them, root soaking > foliar spray > seed soaking. The treatment of seed soaking with quinoa saponins inhibited germination of tomato seeds to some extent. However, the germination rate of tomato seeds after saponin soaking was comparable to the chemical pesticide (thiram carboxin); therefore, this method could be used to control tomato wilt disease. Compared with the antifungal activity in vitro, the antifungal activity of quinoa saponins in vivo was much higher when the saponin concentration was between 0.5–1.0 g/L, indicating that the antifungal activity of quinoa saponins may be achieved mainly by inducing resistance. This investigation supports the potential use of quinoa saponins as a supplier of antifungal compounds, and could be the foundation for a future study examining the use of quinoa bran as a new resource against FOL.

**Author Contributions:** Conceptualization, X.Z. (Xueyong Zhou), H.G., L.Z., L.Y. and Z.W.; methodology, X.Z. (Xueyong Zhou), H.G., Y.N. and X.Z. (Xiaoying Zhang); software, X.Z. (Xueyong Zhou), Y.N., X.Z. (Xiaoying Zhang) and Z.W.; validation, X.Z. (Xueyong Zhou), Y.N. and X.Z. (Xiaoying Zhang); formal analysis, X.Z. (Xiaoying Zhang), H.G., L.Z., L.Y. and Z.W.; investigation, X.Z. (Xueyong Zhou), H.G., Y.N., X.Z. (Xiaoying Zhang) and Z.W.; resources, X.Z. (Xueyong Zhou), H.G., L.Z. and L.Y.; data curation, X.Z. (Xueyong Zhou), Y.N. and X.Z. (Xiaoying Zhang); writing—original draft preparation,

X.Z. (Xueyong Zhou), Y.N. and X.Z. (Xiaoying Zhang); writing—review and editing, X.Z. (Xueyong Zhou), H.G., Y.N. and X.Z. (Xueyong Zhou). All authors have read and agreed to the published version of the manuscript.

**Funding:** This research was funded by the project for Local Science and Technology Development Guided by the Central Government in Shanxi Province, China (YDZJSX2022A055, the Department of Science and Technology of Shanxi Province), the Fundamental Research Program of Shanxi Province, China (202203021221122, the Department of Science and Technology of Shanxi Province), and the Graduate Education Innovation Project of Shanxi Province, China (2023KY470, the Education Department of Shanxi Province).

**Data Availability Statement:** Data are contained within the article.

**Conflicts of Interest:** The authors declare no conflict of interest.

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
