# Peer review of "Crude Saponins from Chenopodium quinoa Willd. Reduce Fusarium Wilt Infection in Tomato Seedlings"

_horticulturae, doi:10.3390/horticulturae9121340_

Round 1

Reviewer 1 Report

Comments and Suggestions for Authors

The use of natural substances derived from plants as plant protection agents against pathogens or plant growth and development stimulators plays an important role in modern plant cultivation technology. Therefore, searching for plant species rich in substances with high fungistatic potential is very justified. In such a context, the article is innovative and has cognitive and utilitarian value. The topic of the article corresponds to its content. I have doubts whether one cycle of research provides the basis for reliable conclusions. The authors do not provide information about the conditions of the experiment. While temperature, humidity, type of substrate (soil), lighting determine the course of pathogenesis, plant development and the effectiveness of protective treatments. Therefore, they cannot be omitted, even if they are in vitro tests. Moreover, I believe that especially chapter 2. Material and methods requires complete completion and clarification.

Detailed comments are included below:

Introduction

Line 83 – what extracts? In this case, it is essential oils that generally have high protective potential. Please write precisely what extracts you mean.

Line 92-94 - but this is just one example of the lack of fungistatic activity in relation to Candida albicans (a yeast) that causes mycosis in humans.

Line 95 – Since quinoa saponins have not been tested, it is reasonable to write about the antifungal effect of saponins found in other plant species. Is that what made you decide to undertake the research?

Material and methods

Line 103 – Is Quinoa Bran an Industrial By-Product? Please write something about it. The use of industrial waste is another pro-ecological aspect of "zero waste".

Line 111 – in 2.2. only the author of the markings was indicated. The methodology for determining saponins should be described in particular detail.

Line 125 - please specify precisely the size of the box, at what distance and at what depth the seeds were placed. There were a total of 200 tomato seeds in each combination?

Line 130 – what plant was FOL isolated from?

Line 144-145 – FOL inoculum on sorghum grain was 3% of the culture medium (soil)?. Please describe carefully.

Line 154-155 -The lab experiment was only performed in three repetitions? I believe there should be a minimum of 5 repetitions.

Line 168 – what was the capacity of the pot?

Line 169 – 35 cuttings x 4 repetitions, i.e. 148 cuttings in one combination? What care treatments were used (watering), what were the temperature, humidity and lighting conditions. The experiments were performed once (in one cycle), so all the breeding parameters should be provided.

Why was the tomato health assessed after 10 days? I understand that the effectiveness of the treatment was assessed at the same time.

  Results

Line 211-212 - Is it about the saponin content in quinoa bran or seeds, or maybe leaves or flowers? – this should be specified (consistently as in the methodology).

Lines 215-217 – what is the deal with the ash content in quinoa saponins? Please specify precisely (seeds, bran).

Line 233 – different superscript letters indicate important differences…

  Discussion

Lines 301-309 – It is worth providing the protective effects (%) of the plant extracts used and comparing them with those obtained in the experiment with quinoa.

The discussion is a confrontation of the literature with the results of your own research - these elements are missing, please refine this part of the manuscript.

Lines 348-350 – So maybe you should consider simply adding quinoa bran to the soil?

  Conclusions

There is a lack of specifics (presentation of effects in numbers, %, etc.

References

Please carefully check the correctness of the citation.

Reviewer 2 Report

Comments and Suggestions for Authors

The manuscript presents the evaluation of a quinoa saponin extract in the prevention of Fusarium infestation in Tomato. The results show a reduction in the effects of Fusarium on seed imbibition and the initial stages of vegetative growth of tomato, which is of importance for the management of tomato seed propagation and in the evaluation of alternative substances for the control of micellar fungi.

The work has an adequate experimental design, including the necessary controls and sufficient replications for the conclusions to be significant.

My suggestion is to reconsider the discussion of the resistance mechanism that can occur in vivo, which is mentioned in the conclusions, since this conclusion is not adequately developed and there is no experimental evidence on the possible mechanism.
